# SAGA Complex Subunit Hfi1 Is Important in the Stress Response and Pathogenesis of *Cryptococcus neoformans*

**DOI:** 10.3390/jof9121198

**Published:** 2023-12-15

**Authors:** Chendi K. Yu, Christina J. Stephenson, Tristan C. Villamor, Taylor G. Dyba, Benjamin L. Schulz, James A. Fraser

**Affiliations:** School of Chemistry & Molecular Biosciences, Australian Infectious Diseases Research Centre, The University of Queensland, Brisbane, QLD 4072, Australia; chendi.yu@uq.net.au (C.K.Y.); christina.stephenson@uq.net.au (C.J.S.); s4359281@student.uq.edu.au (T.C.V.); taylor.dyba@uq.net.au (T.G.D.); b.schulz@uq.edu.au (B.L.S.)

**Keywords:** *Cryptococcus neoformans*, SAGA complex, *HFI1*, *ADA1*, virulence

## Abstract

The Spt-Ada-Gcn Acetyltransferase (SAGA) complex is a highly conserved co-activator found across eukaryotes. It is composed of a number of modules which can vary between species, but all contain the core module. Hfi1 (known as TADA1 in *Homo sapiens*) is one of the proteins that forms the core module, and has been shown to play an important role in maintaining complex structural integrity in both brewer’s yeast and humans. In this study we successfully identified the gene encoding this protein in the important fungal pathogen, *Cryptococcus neoformans*, and named it *HFI1*. The *hfi1*Δ mutant is highly pleiotropic in vitro, influencing phenotypes, ranging from temperature sensitivity and melanin production to caffeine resistance and titan cell morphogenesis. In the absence of Hfi1, the transcription of several other SAGA genes is impacted, as is the acetylation and deubiquination of several histone residues. Importantly, loss of the gene significantly impacts virulence in a murine inhalation model of cryptococcosis. In summary, we have established that Hfi1 modulates multiple pathways that directly affect virulence and survival in *C. neoformans*, and provided deeper insight into the importance of the non-enzymatic components of the SAGA complex.

## 1. Introduction

Adaptation to environmental conditions is required for survival, a process typically achieved via changes in gene expression to produce advantageous phenotypes, such as virulence trait production during infection by pathogenic species. The ability to alter phenotype is influenced by genetic information and environmental factors that impact gene expression. Gene expression is complicated, involving functions ranging from proteins that recruit RNA polymerase, to heritable chemical modifications of DNA and associated proteins that are the foundation of epigenetic phenomena. A widely studied epigenetic modification is the addition of a methyl group to *C*^5^ of cytosine in DNA, which inhibits DNA-binding proteins and plays a role in transcriptional regulation [1,2]. Epigenetic modifications can also occur on the proteins that package DNA, particularly histones, in the form of post-translational modifications (PTMs) [3].

Interactions among different PTMs can be complex, as the enzymes associated with these changes may compete for the same site, or their activity may be influenced by adjacent modifications [4,5]. The enzymes coordinating post-translational modification of histones can be referred to as readers (recognizing PTMs), writers (adding PTMs), or erasers (removing PTMs), and often work as multi-subunit protein complexes. A classic example of readers, writers, and erasers functioning in concert is the Spt-Ada-Gcn Acetyltransferase (SAGA) complex.

SAGA is a co-activator complex that plays a crucial role in regulating gene expression in animals, plants, and fungi. In animals it consists of five modules: HAT (histone acetyltransferase), DUB (deubiquitinase), TF-binding (transcription factor-binding), splicing, and core [6,7,8]. In plants, the proteins of the DUB module exist, but are not associated with SAGA [7]. Likewise, fungi are also missing one of the modules present in animals, only in this case, it is the proteins of the splicing module that are not associated with the complex [9,10].

In *Saccharomyces cerevisiae*, SAGA (ScSAGA) consists of 19 proteins which come together to form the four modules. The TF-binding module of ScSAGA comprises only one protein (Tra1), which interacts with multiple transcription factors such as Gal4, Gcn4, and Hap4 to enable recruitment of the complex [11]. HAT consists of four proteins, namely the histone acetyltransferase Gcn5 (a writer) which acetylates lysine in histones H3 and H2B, Ada2 and Ada3 which promote Gcn5 activity, and Sgf29 (a reader) which recognizes di- and tri-methylated H3K4 to direct HAT activity [12,13,14,15]. The DUB module also includes four proteins, namely the histone deubiquitinase Ubp8 (an eraser) which deubiquitinates histone H2B, Sus1 which contributes to Ubp8 function and links SAGA to mRNA export, Sgf73 which maintains Ubp8 conformation, and Sgf11 which is required for Ubp8 connection to SAGA [16,17,18,19,20].

The largest module within ScSAGA is the core, located at the center of the complex [21,22]. Comprising ten proteins, the core module has five TBP-associated factors (Taf5, Taf6, Taf9, Taf10 and Taf12) that are also components of transcription factor II D (TFIID); formed by these five proteins, eight more Tafs, and TBP in *S. cerevisiae*, TFIID is a multi-protein component of the RNA polymerase II pre-initiation complex [23]. Taf5, Taf6, Taf9, Taf10, and Taf12 are therefore essential. When in ScSAGA, these five Tafs contribute to structural integrity, interacting with basal transcription machinery and transcriptional activators [22,24]. Alongside the five Tafs in the ScSAGA core module are five proteins that are not components of TFIID, namely Hfi1/Ada1, Spt7 and Spt20 (all required for ScSAGA structural integrity), as well as Spt8 and Spt3 (both required for recruitment of TBP) [22].

Together, the complex formed by these modules plays a critical role in globally regulating gene transcription. For example, transcription of *GAL1* (encoding galactokinase) in response to galactose availability involves ScSAGA recruitment through the joint signals of Gal4 interaction with Tra1 (TF-binding module) and Snf1 protein kinase phosphorylation of histone H3 S10 enabling Gcn5 (HAT module) to acetylate histone H3 K9, K14, and K18 to loosen the chromatin structure. Ubp8 (DUB module) de-ubiquitylates histone H2B K123, triggering a cascade of events that sets RNA pol II elongation in motion [25]. ScSAGA also recruits TBP to the *GAL1* TATA box via Spt3 and Spt8 (core module) and interacts with the mRNA export apparatus via Sus1 (DUB module) [26]. These activities are brought together in ScSAGA by the core module.

In *Homo sapiens*, while the molecular functions of each protein in SAGA have not been fully characterized, the complex is known to contain 20 proteins forming five modules; with the exception of the core protein, Spt8, all ScSAGA proteins have homologs that can be found in the human SAGA (HsSAGA) complex [9]. The human core module consists of nine proteins, one fewer than in *S. cerevisiae*; there is no Spt8 homolog.

Investigation of the structure and function of fungal SAGA goes beyond humans and brewer’s yeast to pathogens where it plays a key role in responding to stress caused by the host. These studies typically focus on HAT and DUB. For instance, HAT protein Gcn5 influences cell growth and conidiation in the rice pathogen, *Fusarium fujikuroi*, regulates the high-iron stress response in the cereal crop pathogen, *Fusarium graminearum*, and is required for virulence in the human pathogen, *Cryptococcus neoformans*. The DUB protein, Ubp8, regulates pleiomorphism in the human pathogen, *Candida albicans* and carbon catabolite repression in the rice pathogen, *Magnaporthe oryzae* [27,28,29,30,31]. Studies focusing on the core module are fewer in number, but have shown roles of the core protein, Spt20, in regulating infectious particle formation in *Aspergillus fumigatus*, and Spt7 and Spt8 in invasions by *C. albicans* [32,33].

One of the core module subunit coding genes that has not been studied as extensively in fungi is *HFI1* (Histone 2A Functional Interactor 1, also known as *ADA1* for Alteration/Deficiency in Activation-1). Hfi1 is crucial for maintaining the structural integrity of ScSAGA; in the absence of Hfi1, ScSAGA lacks Taf12, Spt3, Tra1, and the entire HAT module (Gcn5, Ada2, Ada3, and Sgf29) [34]. In comparison, the structure of ScSAGA is maintained in the absence of Spt3 or Spt8, two other proteins from this module [35]. Loss of Hfi1 results in transcriptional defects in a range of genes, and its phenotypic abnormalities are more extensive than occur when the HAT proteins, Ada2, Ngg1/Ada3, or Gcn5, are absent [36]. With the help of Spt7 and Spt20, Hfi1 contributes to the association of DUB with chromatin [37,38]. In HsSAGA, TADA1, which is the homolog of Hfi1, also plays an important role in regard to structure [8]. Given the important role that it plays in humans and brewer’s yeast, it is surprising that Hfi1 has not yet been characterized in a fungal pathogen of humans, where SAGA plays a critical role in virulence.

*C. neoformans* is a widespread opportunistic fungus that poses a significant global health concern; in 2022 it was listed as the top-ranked species in the critical priority group of the WHO fungal priority pathogens list [39]. Commonly found in avian guano, soil, flowers, air, water, and certain tree species [40], infectious particles of *C. neoformans* are inhaled, lodge in the alveoli, and disseminate to and penetrate the blood-brain barrier with the aid of diverse mechanisms such as a polysaccharide capsule, the production of melanin, titan cell formation, and extracellular enzyme secretion [41,42,43,44]. Regulation of these virulence traits and survival in the presence of the stressful environment created by the host requires *C. neoformans* to coordinate transcriptional regulation in response to environmental insult, a task contributed to by SAGA [27,45,46].

The role of HAT (Histone Acetyltransferase) in the regulation of hyper- and hypovirulence in *C. neoformans* is well established [27,45,46]. However, the core module responsible for the association of HAT with the SAGA complex has not been studied. Here, we focused on identifying and characterizing the gene encoding Hfi1 in *C. neoformans*. Deletion of this gene revealed a complex, pleiotropic phenotype that highlighted the functional significance of Hfi1 in virulence traits both in vitro and in vivo, leading to hypovirulence in a murine inhalation model of cryptococcosis. Loss of Hfi1 both influenced the epigenetic architecture of *C. neoformans* through histone modification, and affected the transcription of gene-encoding components of the SAGA complex. Overall, these first studies of Hfi1 in a fungal pathogen revealed the essential role of Hfi1 in mammalian infection by *C. neoformans* and emphasized its significance in regulating virulence traits.

## 2. Materials & Methods

### 2.1. Bioinformatic Analyses

The protein sequence of TADA1 from *H. sapiens* was used to identify the *HFI1* gene in *C. neoformans* strain H99O using tBLASTn analysis [47]. Sequence alignments of Hfi1 (TADA1) from *C. neoformans*, *S. cerevisiae*, and *H. sapiens* were performed using ClustalW v1.4 (MacVector Inc., Apex, NC, USA) and Needleman-Wunsch alignment (EMBOSS Needle, Hinxton, Cambridgeshire, UK).

### 2.2. Media and STRAINS

All strains are listed in Appendix A. Wild-type (*C. neoformans* H99O), *hfi1*Δ and *hfi1*Δ + *HFI1* complemented strains were grown in yeast peptone dextrose (YPD) media (2% bacto-peptone, 1% yeast extract, 2% glucose, 2% agar) at 30 °C unless otherwise stated. Mach1 *Escherichia coli* (Thermo Fisher Scientific, Waltham, MA, USA) strains used for cloning were grown on lysogeny broth (LB) agar (1% tryptone, 0.5% yeast extract, 1% sodium chloride, 2% agar) supplemented with 100 μg/mL ampicillin at 37 °C.

### 2.3. Cloning

The primers and plasmids employed throughout this study are listed in Appendix A. All PCR fragments were generated using Phusion High Fidelity DNA Polymerase (New England Biolabs, Ipswich, MA, USA). To amplify the 5’ and 3’ regions of *HFI1*, *C. neoformans* H99O genomic DNA was used as the template, while *NEO* was amplified from pJAF1 [48]. The PCR products (1 kb upstream of *HFI1*, the *NEO* marker, and 1 kb downstream of *HFI1*) were cloned into pBlueScriptII SK(-) using NEBuilder HiFi DNA Assembly (New England Biolabs) to create pCJS48. The complementation plasmid was used to reintroduce *HFI1* in chromosome 1 [49] by cloning a PCR fragment containing *HFI1* into the Safe Haven locus-targeting plasmid pSDMA25 with NEBuilder HiFi DNA Assembly to create pKY04. All plasmids were sequenced through the Australian Genome Research Facility (AGRF), with resulting chromatograms analyzed using CLC Genomics Workbench 12.1 (QIAGEN, Germany).

### 2.4. Creating HFI1 Gene Deletion and Complementation Strains

To conduct biolistic transformation, the sequenced deletion construct was excised from pCJS48 by digestion with KpnI and SpeI (New England Biolabs) and the complementation plasmid pKY04 was linearized with PacI (New England Biolabs). Biolistic transformation was performed using the W7 hydrochloride modification described by Arras and colleagues [50]. To create the *hfi1*Δ, strain transformants were selected on YPD media supplemented with 100 μg/mL G418. To create the *hfi1*Δ + *HFI1*, complemented strain transformants were selected on YPD media supplemented with 100 μg/mL nourseothricin. Genomic DNA from transformants were prepared via CTAB DNA extraction [51], digested with restriction enzymes, electrophoresed on an agarose gel (1%) overnight at 24 V, and Southern blotted onto Hybond-XL membrane (GE Healthcare, Chicago, IL, USA) [52]. *NEO* and Safe Haven probes were labeled with dCTP [α-32P] (PerkinElmer, Shelton, CT, USA) using the DECAprime II kit (Thermo Fisher Scientific, USA). The blots were hybridized with the *NEO* (*hfi1*Δ) or Safe Haven (*hfi1*Δ + *HFI1*) probes overnight at 65 °C, washed with 0.1% SDS, 2× SSC buffer, and then exposed and developed on Fuji SuperRX medical X-ray film (Fujifilm, Tokyo, Japan) or phosphorimager (Typhoon PLA 9500, GE Healthcare, Chicago, IL, USA).

### 2.5. Phenotype and Virulence Assays

The wild-type, *hfi1*Δ and *hfi1*Δ + *HFI1* strains were grown overnight in YPD liquid media, centrifuged, washed in dH_2_O, then resuspended in 1 mL of dH_2_O and diluted to OD_600_ = 1.0, prior to the preparation of serial tenfold dilutions for each strain. Samples were then spotted as 4 µL aliquots onto a yeast nitrogen base (YNB, 2% glucose, 10 mM ammonium sulfate) media supplemented with compounds of interest (0.5 μg/mL FK506, 0.2 mM NiCl_2_, 5 mM caffeine, 1.5 mM DTT, or 1% BSA). For observation of virulence trait production, 3 μL of 10^−1^ dilution suspension was spotted on the media of interest (Christensen’s urea agar [53], _L_-DOPA agar [54], or egg yolk agar [55]). All traits were tested in triplicate at both 30 °C and 37 °C. Cultures were grown in the dark and the phenotypes were assessed at the 48 h and 72 h time points.

For capsule induction, strains were grown for 24 h and then transferred into RPMI 1640 (Life Technologies, Thermo Fisher Scientific, Waltham, MA, USA) supplemented with 2% glucose, 10% fetal bovine serum (Life Technologies). After 48 h of incubation, cells were stained with India ink (BD Diagnostics, Franklin Lakes, NJ, USA) and imaged using a Leica DM2500 microscope and DFC425C camera (Leica, Wetzlar, Germany). For each strain, ten images were photographed and 100 cells measured for the whole cell and capsule diameters using Adobe Illustrator.

For titan cell induction, approximately 10^7^ cells of each strains were inoculated into 10 mL of liquid YPD media in T25 Nunclon Delta Surface tissue culture flasks (Thermo Scientific) and incubated at 150 rpm for 22 h at 30 °C. A total of 10^5^ cells were collected and titan cells induced in 1 mL of minimal media (15 mM d-glucose, 10 mM MgSO_4_, 29.4 mM KH_2_PO_4_, 13 mM glycine, 3.0 µM thiamine) for 120 h at 30 °C. Cells from each strain were imaged on a hemocytometer at 10× and 40× before and after induction. Cell size and capsule size were measured using Adobe Illustrator and analyzed using Prism 10 (GraphPad Inc., Boston, MA, USA). Significant differences between strains were identified using two-tailed unpaired *t*-tests with Welch’s correction.

### 2.6. Murine Inhalation Model of Virulence

For murine infection assays, 6-week-old female BALB/c mice (Ozgene) were infected by nasal inhalation [56]. For each strain, 10 mice were inoculated with a 50 μL drop containing 5 × 10^5^ *C. neoformans* cells. A maximum of 5 mice were housed per IVC cage (Tecniplast, West Chester, PA, USA) with Bed-o’Cobs 1/8′′ bedding (The Andersons, Maumee, OH, USA), Crink-l′Nest nesting material (The Andersons, USA), and cardboard as environmental enrichment. Mice were provided Rat and Mouse Cubes (Specialty Feeds, Glen Forrest, WA, Australia) and water *ad libitum*. Each mouse was examined and weighed twice daily for the duration of the experiment, with affected mice euthanized via CO_2_ inhalation once body weight had decreased to 80% of their pre-infection weight or they exhibited symptoms consistent with infection. Death after CO_2_ inhalation was confirmed by pedal reflex prior to dissection. The brain, lungs, liver, spleen, and kidneys were collected, homogenized using a TissueLyser II (QIAGEN) and plated to determine colony-forming units per gramme of organ weight; serial dilutions were spotted on YPD agar supplemented with 100 μg/mL ampicillin, 50 μg/mL kanamycin, and 25 μg/mL chloramphenicol to prevent bacterial contamination. Kaplan-Meier survival curves were plotted using GraphPad Prism 10.0 (GraphPad Software, Boston, MA, USA). Significance was analyzed using the log-rank test while organ burden significance was determined using a one-way ANOVA with Tukey’s multiple comparisons test. *p* values of < 0.05 were considered significant. Ethics statement: This study was carried out in strict accordance with the recommendations in the Australian Code of Practice for the Care and Use of Animals for Scientific Purposes by the National Health and Medical Research Council. The protocol was approved by the Molecular Biosciences Animal Ethics Committee of The University of Queensland (AEC approval number: 2022/AE000748). Infection was performed under methoxyflurane anesthesia, and all efforts were made to minimize suffering through adherence to the Guidelines to Promote the Wellbeing of Animals Used for Scientific Purposes as put forward by the National Health and Medical Research Council.

### 2.7. Protein Extraction

Cell pellets were collected via centrifugation and resuspended in 5 mL of tris-buffered saline (TBS, pH 7.5) buffer. Resuspended cells were aliquoted into 1.5 mL screw cap tubes (ThermoFisher Scientific) and fast frozen in liquid nitrogen. A 2X modified lysis buffer (50 mM tris-HCl [pH 7.5], 100 mM NaCl, 100 mM PMSF, and 1× EDTA-free protease inhibitor cocktail [Roche, Basel, Switzerland]) and ~200 µL of 0.5 mm acid-washed silica glass beads (BioSpec Products, Bartlesville, OK, USA) were added, and the cells were homogenized at 4 °C for 90 s with 2 min of rest for 10 cycles on a Tissuelyser II (QIAGEN, Hilden, North Rhine-Westphalia, Germany). After homogenizing, lysate was transferred to Protein LoBind Tubes (Eppendorf, Hamburg, Germany); then glycerol, SDS, DTT, and Triton X-100 were added to final concentrations of 4%, 1%, 1 mM, and 1%, respectively. Lysates were incubated for 10 min at 4 °C on a HulaMixer Sample Mixer (ThermoFisher Scientific), and then centrifuged at 14,000 rpm at 4 °C for 20 min. The supernatant was collected as the protein extract.

### 2.8. Western Blot Analyses

Protein concentrations were measured using the Pierce Detergent Compatible Bradford Assay Kit (ThermoFisher Scientific). A total of 15 ng of protein extracts were electrophoresed at 80 V for 3 h on 12% sodium dodecyl sulfate (SDS)-polyacrylamide gels, then transferred to Hybond-P PVDF membrane (GE Healthcare, USA). For detecting histone modification, a 1:5000 dilution of anti-H2B (#12364), anti-H2BK120ub (#5546), anti-H3 (#4499), anti-H3K9ac (#9649), anti-H3K14ac (#7627), anti-H3K18ac (#13998), anti-H4K8 (#2594), anti-H4K12 (#13944), anti-H4K16 (#13534), or anti-H3K4trimethl (#9751) rabbit monoclonal antibody (Cell Signaling Technology, Danvers, MA, USA) was used. All primary antibodies were diluted in TBST (25 mM tris-HCl [pH 7.5], 100 mM NaCl, 0.1% Tween 20) with 5% bovine serum albumin and incubated overnight at 4 °C. Goat anti-rabbit IgG, HRP-linked antibody (Cell Signaling Technology, USA, #7074) was used as a secondary antibody and diluted 1:1000 in TBST. After incubation with the secondary antibody for 1 h at room temperature, protein was detected using the SuperSignal West Pico PLUS Chemiluminescent Substrate Kit (ThermoFisher Scientific) and an ImageQuant 800 GxP biomolecular imager (Cytiva Life Sciences, Marlborough, MA, USA).

### 2.9. RNA Extraction and Reverse Transcription-Quantitative PCR

All primers used are listed in Appendix A. Strains were grown in YPD at 30 °C until OD_600_ = 1.0. Cultures were harvested and cell pellets were frozen, which were then lyophilized. Total RNA was isolated from wild-type and the *hfi1*Δ mutant using TRIzol reagent (Invitrogen), and cDNA generated using the Superscript III First-Strand Synthesis System (Invitrogen). Primers for SAGA genes were designed to span exon–exon boundaries. SYBR Green supermix (Applied Biosystems, Waltham, MA, USA) and an Applied Biosystems ViiA 7 Real-Time PCR System were used for performing quantitative real-time PCR (qRT-PCR). The relative gene expression was quantified using the change-in-threshold (2^−ΔΔCT^) method [57]. Four house-keeping genes, *ACT1*, *GPD1*, *HHT1*, and *TUB2*, were used as controls for normalization; calculations were performed using *ACT1* as the control. To determine statistical significance, a one-way analysis of variance (ANOVA) was performed using the unpaired with Tukey’s multiple comparisons test in GraphPad Prism Version 10.0.

## 3. Results

### 3.1. Identification of the Gene-Encoding Hfi1 in C. neoformans

Previous bioinformatic analyses investigating SAGA in *C. neoformans* identified many of the components of the complex, but not Hfi1 [27]. To identify the gene-encoding component of the SAGA core module, we performed a tBLASTn analysis of the genome of type strain H99O using TADA1 from the host species *Homo sapiens*. *CNAG_06852*, which we have named *HFI1*, was the only hit identified. Pairwise comparisons between the *C. neoformans*, *S. cerevisiae*, and *H. sapiens* proteins revealed that this component of SAGA has a low level of conservation (Appendix A); the *C. neoformans* protein exhibits 11.7% identity (21% similarity) to its *H. sapiens* counterpart, *S. cerevisiae* displays 14.8% identity (25.9% similarity) to *H. sapiens*, and *C. neoformans* shows 20% identity (33% similarity) to *S. cerevisiae.*

The annotation of *CNAG_06852* in the H99O genome is based on transcriptomic data, providing a robust model for the structure of *HFI1*. The gene has a transcript of 1569 bp and has two introns within an ORF that encodes a 522 residue protein.

*HFI1* was deleted from strain H99O using biolistic transformation, replacing the ORF with the *NEO* selectable marker [48]. Following validation by Southern blot, the mutant was complemented by the insertion of a wild-type allele at Safe Haven 1, a well-characterized gene-free neutral site on chromosome 1 [49].

### 3.2. Hfi1 Is Essential for C. neoformans Growth during Stress

With the availability of the *hfi1*Δ mutant and its complemented derivative (*hfi1*Δ + *HFI1*), the consequences of the loss of Hfi1 from the core module could be investigated. In *S. cerevisiae*, the *hfi1*Δ mutant behaved as wild-type on normal synthetic complete media, but was sensitive to caffeine [35]. In this study, we compared the *hfi1*Δ mutant with the wild-type and the complemented strain *hfi1*Δ + *HFI1* strains on nutrient-rich YPD and synthetic complete YNB. While the strains were indistinguishable at 30 °C, the *hfi1*Δ strain exhibited impaired growth under the stressful condition of human body temperature (37 °C) on both YPD and YNB media (Figure 1).

The *hfi1*Δ mutant does not exhibit a consistent response to all stresses (Figure 1). Growth was inhibited by the presence of NiCl_2_ and oxidative stress provoked by dithiothreitol (DTT), even at 30 °C. In contrast, the mutant grows more strongly than wild-type or the complemented strain under cell wall stress induced by caffeine, the opposite phenotype displayed by the *S. cerevisiae hfi1*Δ mutant on this media. Unexpectedly, we observed an unusual phenotype with the mutant when in the presence of the calcineurin inhibitor, FK506 (tacrolimus): the mutant exhibits sensitivity to the compound at 30 °C, but has increased drug tolerance compared to wild-type at 37 °C.

### 3.3. Deletion of HFI1 Impacts C. neoformans Virulence Traits In Vitro

Beyond general stress responses, deletion of *HFI1* also influences several virulence traits that are important for *C. neoformans* during the infection process. Loss of this component of the core module results in reduced production of melanin at both 30 and 37 °C on l-3,4-dihydroxyphenylalanine (_L_-DOPA) agar, less production of phospholipase B at 37 °C on egg yolk agar, and reduced extracellular protease production at both 30 and 37 °C on BSA media (Figure 2). The phenotype of the *hfi1*Δ mutant on Christensen’s agar is twofold: first, as shown by the stronger than wild-type orange halo, the *hfi1*Δ mutant has increased urease activity at 37 °C; second, its growth on this vitamin-free media indicates that the *C. neoformans* mutant is not a myo-inositol auxotroph, which is a phenotype present in its *S. cerevisiae* counterpart [35] (Figure 2).

One distinctive characteristic of *Cryptococcus* among fungal pathogens lies in its polysaccharide capsule, which plays a role in countering the host immune system. Investigation of capsule formation using India ink staining following growth in RPMI 1640 medium revealed that the *hfi1*Δ strain has a smaller capsule than wild-type (Figure 3A). Significant differences (*p*-value < 0.0001) were observed between the *hfi1*Δ strain and the wild-type at both 30 °C and 37 °C (Figure 3B).

To prevent phagocytic engulfment, *C. neoformans* cells can increase their physical size to create titan cells while in the host lung [58], an unusual morphotype that allows *C. neoformans* to change the host immune response and increases antifungal drug resistance [59]. The titan cell phenotype is detectable in vitro following incubation of the pathogen in minimal medium at 30 °C for five days [58]. Intriguingly, the *hfi1*Δ strain failed to induce titan cell production, and this phenotype was rescued by reintroduction of the wild-type allele (Figure 4A). To further support the observation, measurement of 300 cells’ diameter (excluding capsule) for each strain followed by a statistical analysis were performed. Significant differences (*p*-value < 0.0001) were observed between the *hfi1*Δ strain and the wild-type (Figure 4B).

### 3.4. HFI1 Is Important for C. neoformans Virulence in a Murine Inhalation Model of Infection

To assess the impact of deleting the *HFI1* gene in a murine inhalation model of infection, we investigated whether the absence of this core module gene in SAGA affects the virulence of *C. neoformans* in vivo. Given that Hfi1 plays an important role in SAGA structure in other species, it was predicted that the deletion of *HFI1* would result in a reduction in virulence.

Groups of ten mice were infected with wild-type, the deletion mutant, or the complemented strain. Daily monitoring of the mice post-infection was carried out, and euthanasia criteria were the manifestation symptoms of meningoencephalitis or a 20% reduction in body weight. After euthanasia, the spleen, kidney, liver, lungs, and brains from all mice were collected, homogenized, and cultured on YPD media to quantify the fungal burden.

By the middle of week 5 all mice infected with wild-type H99O and *hfi1*Δ + *HFI1* strains succumbed to illness. This outcome was corroborated by equivalent fungal burden loads across all organs in mice infected with wild-type and the complemented strains (*p*-value > 0.05). In contrast, mice infected with the *hfi1*Δ mutant not only survived, but appeared healthy and consistently gained weight during the experiment (Figure 5A and Appendix A). Upon termination of the experiment at day 50, while no fungal burden was detected in the homogenized liver, spleen, and kidney of mice infected with the mutant, colonies were identified in the lungs of seven mice, with two mice also showing colonies in the brain (Figure 5B). Loss of *HFI1* leads to a reduction in virulence in *C. neoformans*, conclusively establishing the crucial role of *HFI1* in *C. neoformans* during the infection process.

### 3.5. Hfi1 Is Required for SAGA to Perform a Subset of Its Post-Transcriptional Modification Functions

A fundamental role shown for both ScSAGA and HsSAGA is the acetylation of various lysine residues on histones, such as H3K9, H3K14, H3K18, and other lysines present in H4, such as H4K8, H4K12, and H4K16 [60,61,62,63,64]. These acetylation activities are influenced by the tri-methylation of H3K4, which is recognized by HAT module protein, Sgf29 [65,66]. ScSAGA and HsSAGA also mediate the de-ubiquitination of H2B lysine residue, K120 for human or K123 for the brewer’s yeast [67,68]. To investigate whether Hfi1 is required to be in the core module for SAGA modification of histones to occur, the post-translational modification of histones was investigated in wild-type, *hfi1*Δ and *hfi1*Δ + *HFI1* strains using Western blot analyses (Figure 6).

The deletion of *HFI1* resulted in a higher level of ubiquitination of *C. neoformans* H2BK133 (*S. cerevisiae* H2BK123, *H. sapiens* H2BK120), suggesting that the loss of this core protein impacts the ability of the DUB module to function (Figure 6A). Investigation of histone acetylation revealed a reduction in acetylation of H3K14 and, to a lesser extent, K18, but no apparent influence on acetylation of lysine residues in histone H4 (Figure 6B,C). No impact of trimethylation of H3K4 was observed (Figure 6C). Overall, the loss of Hfi1 did impact a number of histone PTMs, but did not lead to the complete abolishment of any of those investigated (Figure 6D).

### 3.6. Deletion of Hfi1 Influences the Expression of Other SAGA Genes

In order to gain deeper insight into the mechanisms underlying the impact of the loss of Hfi1 on histone modifications, we conducted an analysis of the transcript levels of all genes predicted to encode a component of the SAGA complex, comparing transcription between wild-type and the *hfi1*Δ mutant using RT-qPCR. Interestingly, the deletion of *HFI1* resulted in the induction of expression of some SAGA genes (Figure 7); the abundance of *SPT3*, *SUS1*, and *TAF10* transcripts was significantly increased (*p*-value < 0.0001), while the abundance of *SPT20*, *TAF5*, and *TAF12* transcripts was slightly increased (*p*-value < 0.01). Importantly, the abundance of the histone acetylase-encoding *GCN5* or the deubiquitinase-encoding *UBP8* genes was not affected, indicating that the effect of histone PTMs in the *hfi1*Δ mutant is not through loss of the transcription of these genes.

## 4. Discussion

SAGA is an evolutionarily conserved complex that functions as a transcriptional regulator of stress-response genes. In *S. cerevisiae*, SAGA is well characterized and can be divided into four modules, namely HAT, DUB, TF-binding, and core which binds all the other modules together. In *H. sapiens*, the SAGA complex contains a splicing module. Until now, studies in *C. neoformans* have been limited to the HAT module. In this study, we began to investigate the core module in *C. neoformans*, starting with Hfi1. The understanding of this protein in fungi is extremely limited; the *S. cerevisiae hfi1*Δ mutant exhibits sensitivity to caffeine, a requirement for inositol [35], while in the soil-borne plant pathogen, *Verticillium dahliae*, the mutant has defects in multiple phenotypes contributing to virulence [69]. In this study, we also observed a pleiotropic phenotype for the *hfi1*Δ mutant, this time in the top-ranked Critical Priority Group species from the World Health Organization fungal priority pathogens list.

The phenotypes observed in *C. neoformans* differed from those observed in *S. cerevisiae* and *V. dahliae*, but nevertheless shared a common theme. The *C. neoformans hfi1*Δ mutant exhibited sensitivity to stress introduced by certain metals (NiCl_2_), reducing agents (DTT), cell wall stressors (caffeine), and anti-fungal drugs (FK506). Focusing on phenotypes directly associated with the infection process, the mutant displayed weaker growth at human body temperature, a smaller capsule, no titan cell formation, less melanin, less protease, less phospholipase B, and higher urease production. When challenged in a mouse inhalation model of infection, the mutant was unable to cause detectable disease even after 50 days, with 30% of the individuals completely clearing the infection, and the remaining mice continuing to thrive and gain weight. These findings lead to the conclusion that the loss of *HFI1* decreases virulence but may not abolish it completely. Previous studies in *C. neoformans* have shown that a loss of *GCN5* and *ADA2* both lead to avirulence; however, these studies did not include an investigation of whether fungus was present in mice that were still alive at the conclusion of the experiment. In contrast, deletion of *SGF29* results in hypervirulence [27,45,46]. Such divergent results within the same complex require further investigation.

The in vivo infection result shows that Hfi1 is required for *C. neoformans* to exhibit wild-type virulence, and the extensive in vitro phenotypes provide insight into the reason why this is the case. Multiple well-established virulence traits were compromised due to the absence of this protein. The molecular mechanism underpinning these phenotypes is likely the major function of SAGA modifying PTMs becoming compromised.

In *S. cerevisiae*, Gcn5 acetylates histone lysine residues, H3K9, H3K14, H3K18, H4K8, H4K12, and H4K16, Sgf29 recognizes H3K4 tri- and di- methylation, and Ubp8 deubiquitinates histone lysine residue, H2BK123 [14,60,70]. In *C. neoformans*, the *gcn5*Δ mutant has been shown to exhibit a reduced acetylation level of H3K9, K14, K18, and K27 [71]. However, the influence of the deletion of SAGA genes on the acetylation of histone, H4, or deubiquitination of histone, H2B, has not previously been studied. With the aid of an analysis spanning eight PTMs, it was shown that Hfi1 influences H3K14 and H3K18 acetylation and H2B deubiquitination. This impact on histone PTMs aligns with similar analyses in *V. dahliae*, and suggests the essential role of Hfi1 in maintaining the integrity of SAGA [69]. A full understanding of the role of the SAGA cotranscriptional activator in the pathogenesis of *C. neoformans* requires further analyses of this important structural element of the complex.

## Figures and Tables

**Figure 1 jof-09-01198-f001:**
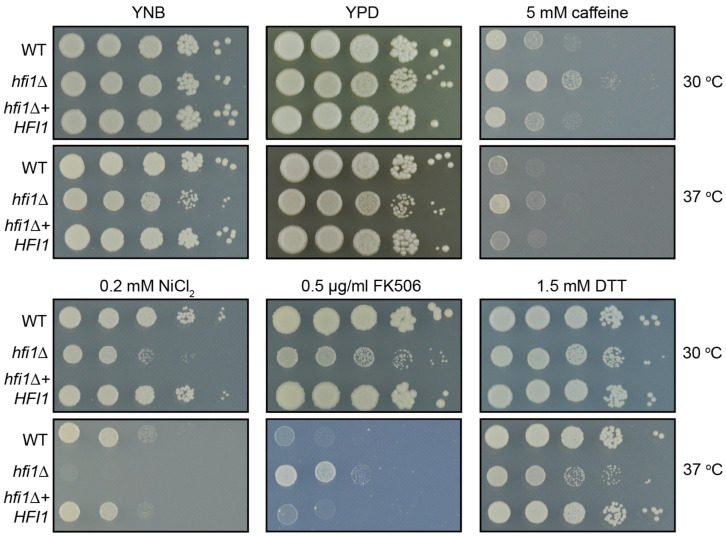
**Loss of *HFI1* results in different responses to stresses.** Growth of 10-fold serial dilutions of wild-type (WT), *hfi1*Δ, and *hfi1*Δ + *HFI1* strains of *C. neoformans* on a variety of media. Pictures were taken after 48 h of growth.

**Figure 2 jof-09-01198-f002:**
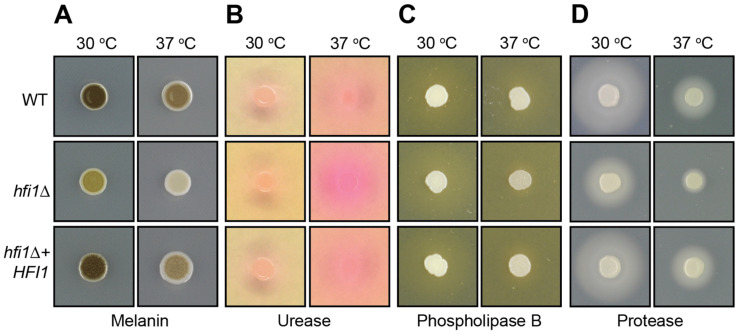
**Loss of *HFI1* affects the production of *C. neoformans* virulence factors in vitro.** (**A**) Melanin production was determined on _L_-DOPA agar, (**B**) urease production on Christensen’s urea agar, (**C**) production of phospholipase B on egg yolk agar, and (**D**) protease production on 2% glucose YNB agar supplemented with 0.1% bovine serum albumin.

**Figure 3 jof-09-01198-f003:**
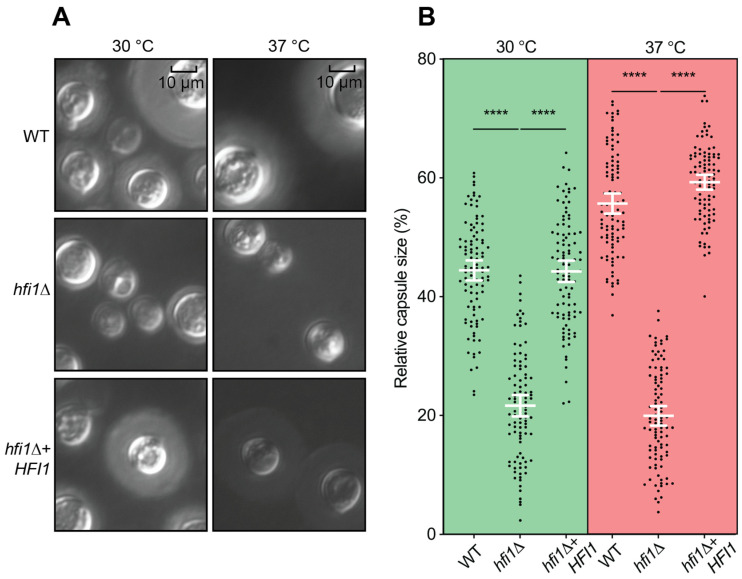
**The *hfi1*Δ mutant has a capsule defect.** (**A**) Strains were incubated in RPMI 1640 medium supplied with 10% fetal bovine serum at 30 and 37 °C. At 48 h, cells were stained with India ink. (**B**) Relative capsule size of 100 cells from wild-type, *hfi1*Δ and *hfi1*Δ + *HFI1*. Strains were compared by two-tailed *t* tests (unpaired) with Welch’s correction. Error bars show the means ±95% confidence intervals, *n* = 100. **** represents significant difference (*p* < 0.0001).

**Figure 4 jof-09-01198-f004:**
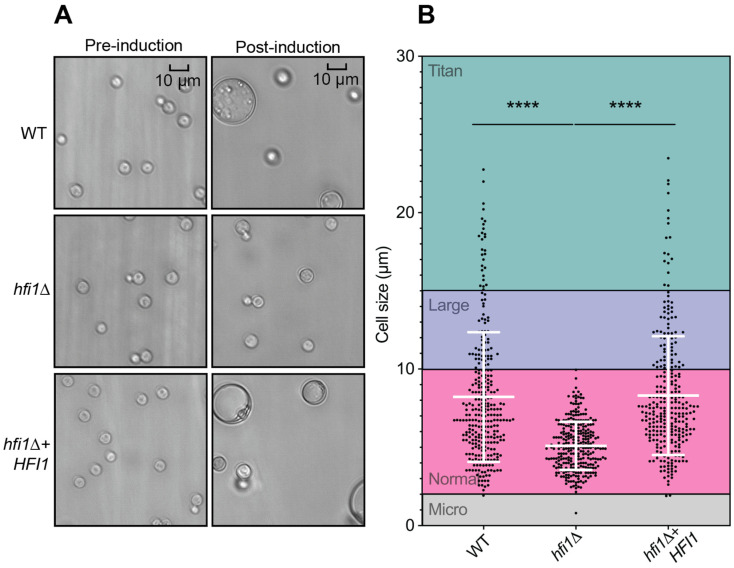
**The *hfi1*Δ mutant does not form titan cells.** (**A**) 10^5^ cell number for each strain were incubated in MM^T^ medium at 30 °C. Images were captured before and after 5 days of incubation. (**B**) Cell sizes (without capsule) of 300 cells from wild-type, *hfi1*Δ and *hfi1*Δ + *HFI1*. Strains were compared by two-tailed *t* tests (unpaired) with Welch’s correction. Error bars show the means ±95% confidence intervals, *n* = 300. **** represents significant difference (*p* < 0.0001).

**Figure 5 jof-09-01198-f005:**
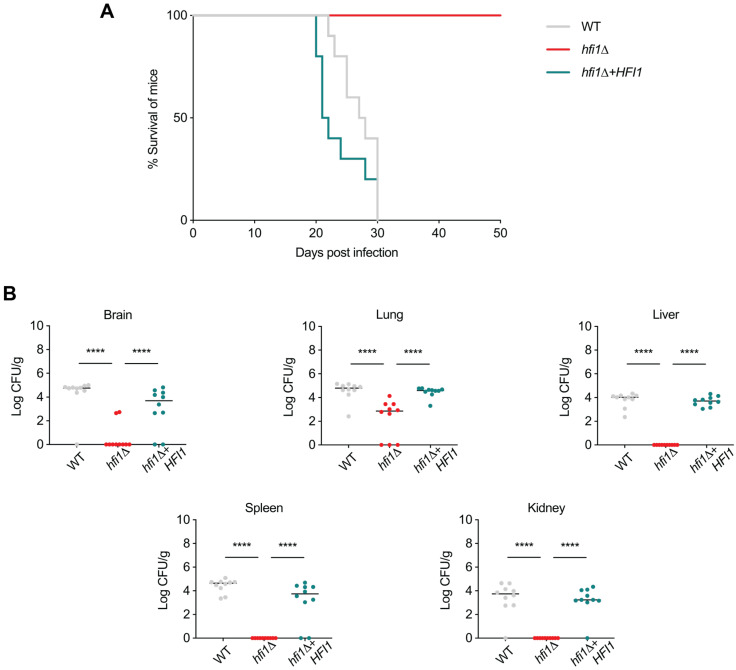
***HFI1* is required for virulence of *C. neoformans* in a murine inhalation model of infection.** (**A**) Five-week-old mice were subjected to intranasal infection with either 10^5^ cells of the wild-type strain, the *hfi1*Δ mutant strain, or the *hfi1*Δ + *HFI1* complemented strains. Kaplan–Meier survival curves were generated, and statistical significance was determined using log-rank tests. All mice infected with the *hfi1*Δ mutant strain survived, and there was no statistically significant difference in survival between mice infected with the wild-type strain and those infected with the *hfi1*Δ + *HFI1* complemented strains (*p*-value > 0.05). (**B**) Fungal organ burden in the infected mice was assessed for wild-type, *hfi1*Δ mutant, and *hfi1*Δ + *HFI1* complemented strains. Statistical significance was determined using one-way ANOVA with Tukey’s multiple comparisons test. *p*-values of < 0.05 were considered significant. Median is represented by a black line. Fungal burden in mice infected with the wild-type strain and *hfi1*Δ + *HFI1* complemented strain was significantly higher (**** *p*-value < 0.05) compared to mice infected with the *hfi1*Δ mutant strain.

**Figure 6 jof-09-01198-f006:**
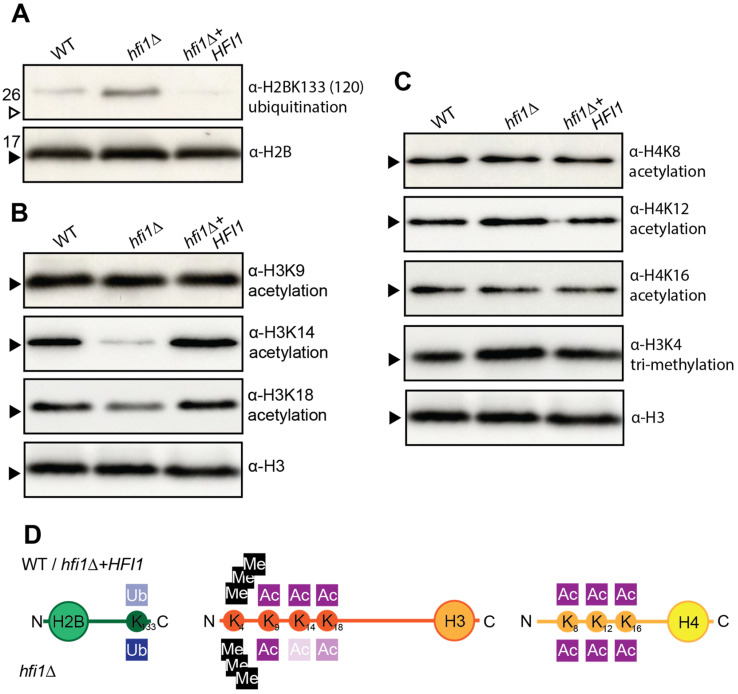
**Hfi1 influences histone deubiquitination and acetylation in *C. neoformans*.** (**A**) Ubiquitination of histone 2B was detected using anti-H2BK120ub antibody. (**B**) Acetylation of histone H3 was detected using anti-H3K9ac, -H3K14ac, and -H3K18ac antibodies. (**C**) Acetylation of histone H4 and methylation of H3 was detected using anti-H4K8ac, -H4K12ac, -H4K16ac, and -H3K4trimethyl antibodies. Arrows indicate the 17 kDa (black arrow) and 26 kDa (white arrow) bands in the Broad Range Color Prestained Protein Standard (NEB). (**D**) Summary of Western blot results.

**Figure 7 jof-09-01198-f007:**
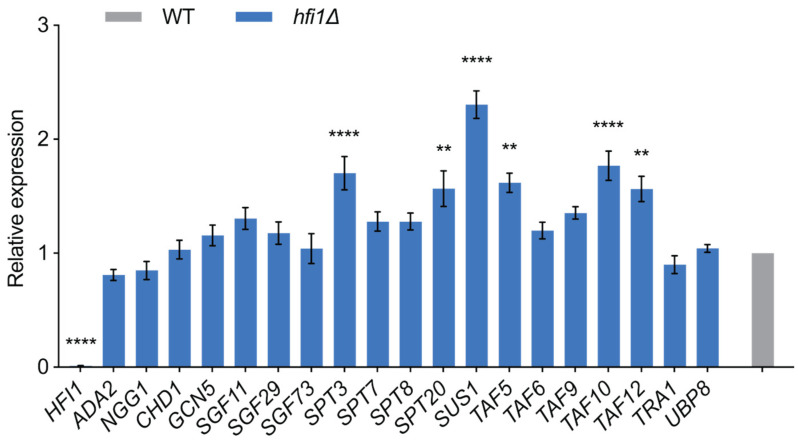
**Loss of *hfi1*Δ influences the abundance of the transcripts of a number of SAGA protein-encoding genes.** *ACT1* was used as the control for normalization. Error bars illustrate the standard errors computed from three biological replicates, each with three independent technical replicates, while the asterisks indicate significant differences (**** *p*-value < 0.0001; ** *p*-value < 0.01) in expression relative to wild-type, as determined by one-way ANOVA with Tukey’s multiple comparisons test.

## Data Availability

All data described in this study are presented in the manuscript and publicly available.

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
