# Peer review of "SAGA Complex Subunit Hfi1 Is Important in the Stress Response and Pathogenesis of *Cryptococcus neoformans"

_jof, 2023, doi:10.3390/jof9121198_

Round 1
Reviewer 1 Report
Comments and Suggestions for Authors
The aim of this study is to establish that Hfi1 modulates multiple pathways that directly affect C. neoformans virulence and survival, and to investigate the importance of non-enzymatic components of the SAGA complex. The article is well written and the bibliography is appropriate, although more recent references could be added.
Introduction
Line 90: isn't it possible to use a more recent reference? Ref 27 ?
Line 110-111 : Today, the biotopes of Cryptococcus yeasts are more extensive than trees and bird droppings. Please add more recent references.
Line 114 : Add more recent references on virulence factors
Line 136 : the reference strain used I the study was H99, It is written H99O ?
Line 187 : Titan cells is mentioned. There's no information on the titans in the introduction.
The results are well presented and convincing. The discussion is short but effective.
Author Response
Response to comment 1: Unfortunately, ref 27 is the most recent paper about the influence of GCN5 in the virulence of Cryptococcus neoformans. The 2023 paper (https://doi.org/10.1128/msphere.00299-23) is about GCN5 in C. neoformans but in a different scope (mating and sexual reproduction, not the virulence).
Response to comment 2: Updated, Please see the attachment.
Response to comment 3: Updated, Please see the attachment.
Response to comment 4: Thanks for mentioning that, we used H99O in this study, which belongs to H99. The previous paper (10.1038/s41598-017-18106-2) describes H99 is a complex, which includes H99O, H99L, H99W etc.. We used H99O in the description just want to make things clear.
Response to comment 5: Updated, Please see the attachment.

Reviewer 2 Report
Comments and Suggestions for Authors
The article by Yu et al describes that a SAGA complex sub-unit (Hfi1) regulates several pathways in Cryptococcus neoformans that are associated with stress response and pathogenesis. The article is very well written and provides strong datasets supporting the conclusion. Here are my comments:
1)Since the authors used 4 housekeeping gene during qPCR analysis, did the expression of any of the housekeeping genes vary in the Hfi1 mutant compared to the WT? Please clarify.
2)Does the expression of HFI1 vary in the WT in response to the stress agents used in the study? For example, in the WT does the expression of HFI1 vary in response to 0.2mM NiCl2? Please clarify.
3)Is the expression of HFI1 temperature sensitive (30Cvs37C)? Please clarify.
4)During protein analysis, what temperature the cells were grown? Will the protein expression differ in the mutant vs WT at 30Cvs37C? Please clarify.
Author Response
Response to question 1: No, according to the raw data, none of the housekeeping genes varied in the Hfi1 mutant compared to the WT. The reason for using four housekeeping genes in this study is just in case, one or some of the housekeeping genes varied between the mutant and WT. Since all four housekeeping genes are not varied, we chose a commonly used housekeeping gene, ACT1 for displaying the data.
Response to questions 2 and 3: The purpose of RT-qRCR analysis we did in this study was to check whether loss of the HFI1 gene will influence the expression of other SAGA genes, similar analysis was conducted in a recent study (https://doi.org/10.3389/fmicb.2022.852571) which we followed closely. In our study, strains used in this analysis were all grown in normal conditions (YPD at 30 degrees Celsius).
Testing the expression of HFI1 in other stressful conditions such as 0.2mM NiCl2 or 37 degrees Celsius was beyond the purpose and scope of this analysis.
Response to question 4: The main function of the SAGA complex is modifying histone markers, such as acetylation, and ubiquitination. The purpose of the protein analysis (Western blot) is to analyse whether loss of HFI1 gene, and therefore loss of Hfi1 protein which is a subunit of the SAGA complex, will influence the level of different histone modifications. A similar analysis was conducted in a recent study (https://doi.org/10.3389/fmicb.2022.852571) which we followed closely. All the antibodies used in the Western blots were specifically targeting these histone markers indicated. Hence, in this set of experiments, all strains were grown in normal conditions (YPD at 30 degrees Celsius).
Testing the expression of whole protein was not the purpose of this specific analysis.